behaviour, ecology, evolution

current–future trade-offs, life-history theory, matching, provisioning rules, quality-quantity trade-offs, residual reproductive value

**Author for correspondence:**
Aisha C. Bründl
e-mail: aisha.bruendl@gmail.com

# Experimentally induced increases in fecundity lead to greater nestling care in blue tits

Aisha C. Bründl[1,2], Enrico Sorato[2], Louis Sallé[2], Alice C. Thiney[2], Sonja Kaulbarsch[1], Alexis S. Chaine[2,3] and Andrew F. Russell[1]

[1]Centre for Ecology and Conservation, University of Exeter, Penryn Campus, Treliever Road, Penryn, Cornwall TR10 9FE, UK
[2]Station d'Ecologie Théorique et Expérimentale (UMR5321), CNRS, Université Paul Sabatier, 2 route du CNRS, 09200 Moulis, France
[3]Toulouse School of Economics, Institute for Advanced Studies in Toulouse, 21 allée de Brienne, 31015 Toulouse, France

ACB, 0000-0001-9887-3229

Models on the evolution of bi-parental care typically assume that maternal investment in offspring production is fixed and predict subsequent contributions to offspring care by the pair are stabilized by partial compensation. While experimental tests of this prediction are supportive, exceptions are commonplace. Using wild blue tits (*Cyanistes caeruleus*), we provide, to our knowledge, the first investigation into the effects of increasing maternal investment in offspring production for subsequent contributions to nestling provisioning by mothers and male partners. Females that were induced to lay two extra eggs provisioned nestlings 43% more frequently than controls, despite clutch size being made comparable between treatment groups at the onset of incubation. Further, experimental males did not significantly reduce provisioning rates as expected by partial compensation, and if anything contributed slightly (9%) more than controls. Finally, nestlings were significantly heavier in experimental nests compared with controls, suggesting that the 22% average increase in provisioning rates by experimental pairs was beneficial. Our results have potential implications for our understanding of provisioning rules, the maintenance of bi-parental care and the timescale over which current–future life-history trade-offs operate. We recommend greater consideration of female investment at the egg stage to more fully understand the evolutionary dynamics of bi-parental care.

## 1. Introduction

Bi-parental care, where offspring are reared jointly by their mother and putative father, is widespread in the animal kingdom and is the norm for birds [1,2]. The challenge with explaining the evolution of such systems revolves around stabilizing the joint contribution of unrelated partners to costly care behaviours, because each member of the pair will benefit from the other contributing more than its 'fair' share [3,4]. Traditional theory predicts that bi-parental care is stabilized when decreases in contributions by one member of the pair are met with only partial compensatory increases by the other [5–7]. This is because, under such response rules, the 'cheating' member of the pair will typically lose more from reducing its contribution than it gains. While meta-analyses confirm that incomplete compensation is a usual response to partner manipulation of provisioning rates [8], frequent empirical exceptions and more recent theory suggest that deviations from the classic expectation of partial compensatory responses can arise when assumptions made in traditional models are relaxed [9,10].

Classic bi-parental care models assume that contributions to offspring production (hereafter prenatal investment) are fixed genetically or by underlying

differences in individual quality (e.g. [5–7]). However, it is now recognized that mothers can allocate their finite resources differentially—increasing or decreasing prenatal investment according to the 'anticipated' relative lifetime fitness returns arising from a current reproductive attempt [11,12]. Given that investment in offspring production can be costly [13–16], changing prenatal investment should impact a mother's subsequent contributions to offspring care following birth/hatching (hereafter postnatal investment) [9]. However, the nature of the relationship between pre- and postnatal investment will depend on the manifestation of current–future trade-offs [3,17,18]. For example, if such trade-offs operate between prenatal and postnatal investment within reproductive events, as has been recently suggested (e.g. [14,19,20]), then we would predict higher investment into offspring production to be associated with a relatively lower postnatal investment in offspring [9]. By contrast, if trade-offs primarily operate *between* breeding events (i.e. between current and future offspring), we would predict that increased prenatal investment will be associated with either no change or relative increases in postnatal investment in the current breeding attempt (i.e. more positive prenatal–postnatal investment associations). This latter prediction arises because the increasing costly prenatal investment will reduce the residual value of future reproductive events (see above), thus favouring increased overall investment in the current attempt. Despite these contrasting predictions, owing to the timescale over which current–future life-history trade-offs are expected to operate, the consequences of variation in prenatal investment for maternal contributions to postnatal offspring care remain to be tested directly.

In turn, in bi-parental care systems, a female's prenatal investment strategy and its impacts on her subsequent level of postnatal care might be expected to change the optimal response rule of her male partner from partial compensation. For example, in a game-theoretic model, Savage *et al.* [9] suggested that male partners can benefit from fully compensating any reduction in maternal postnatal care that results from trade-offs with increased prenatal maternal investment, because high prenatal investment increases overall brood value (see also [21]). In another such model, Johnstone & Hinde [10] showed that when the female is more informed about the value of a current brood, which might be expected if the value is linked to prenatal investment, then the optimal male partner response rule can shift from partial compensation, through no compensation to matched responses. Further, the extent of this shift was shown to depend on the ratio of variation in brood value relative to variation in personal state: compensation is expected when this ratio is in favour of the parent; matching is expected when the reverse is true; while no compensation is expected when this ratio is balanced between parent and offspring. Either way, differential prenatal investment by females is likely to impact the response rules of male partners, but again direct tests of this hypothesis are lacking.

Here we test experimentally the impacts of prenatal investment on levels of postnatal care by females and their partners in the bi-parental blue tit (*Cyanistes caeruleus*). Female blue tits build the nest and incubate the eggs alone, but both members of the pair jointly provision the dependent young with invertebrate prey. We increased prenatal investment by inducing female wild blue tits to lay more eggs, but ensured they neither

incubated nor reared more young; so removing confounding effects of incubation investment and post-hatching brood size on subsequent contributions to nestling provisioning and brood mass. Our study is based on a central tenet of life-history theory—that current–future trade-offs underpin optimal reproductive investment in iteroparous organisms [17,22]. First, if this trade-off operates between phases within reproductive events, experimental females will show lower provisioning rates than controls. By contrast, if it primarily operates among reproductive events, experimental females will show increased provisioning rates relative to controls. Second, classic partner responses rules will be supported if males show patterns of provisioning opposite to that of females in experimental versus control nests, but other responses are possible if males extract information from maternal provisioning patterns and net fitness is incremented by investing more in valuable broods [9,10].

## 2. Material and methods

We performed our study over two consecutive breeding seasons (April–June) in 2013–2014 in a colour-ringed nest-box population of blue tits located within 15 km of the Station for Theoretical and Experimental Ecology in Moulis (42°57′29″ N, 1°05′12″ E) in the French Pyrenees. The Woodcrete boxes ($n =$ approx. 600 Schwegler 2 M with 32 mm entrance diameter; Schorndorf, Germany) are positioned at approximately 50 m intervals in woodlots of mixed deciduous woodland across a 1000 m altitudinal gradient. However, occupancy of high elevation nest-boxes is relatively low, and the vast majority of nests (92%) used in this study were from relatively low elevation woodlots (elevation of nest-boxes used in this study $= 618 \pm 156$ m (mean $\pm$ s.d.), range: 461–1105 m). The woodlots are positioned within expansive woodland networks of the French Pyrenees, which might explain why only 21% of birds ($n = 70$ ringed adults) used in this study bred in nest-boxes in the following years and why only two ringed females breeding in both 2013 and 2014 were included; with each being subjected to the opposite treatment in the second year to minimize effects of pseudo-replication on parental care. All other females were found only once, in the breeding seasons of either 2013 or 2014.

### (a) Experimental design

Blue tit nests were identified during nest building. The date on which the first egg was laid in each nest was known with precision owing to daily checks from when nests neared completion. Experimental ($n = 34$) and control ($n = 16$) nests were assigned at random when at least two nests within 300 m distance overlapped in lay date (maximum two-day difference). Doing so ensured that there was no systematic difference in lay date between experimental (mean: 13th April, $\pm 5$ d (s.d.)) and control nests (mean: 13th April, $\pm 8$ d (s.d.)); Welch's *t*-test ($n = 34,16$): $t_{19.49} = 0.058$, $p = 0.95$). 'Pairing' nests spatially also ensured that we minimized any differences between control and experimental nests in habitat quality and altitude (experimental mean altitude ($\pm$ s.d.): $636 \pm 165.9$ m; control mean: $580 \pm 131.0$ m; Welch's *t*-test ($n = 34,16$): $t_{36.70} = 1.28$, $p = 0.21$). Overall, control nests were visited with similar regularity as experimental nests to monitor egg-laying and obtain the precise dates of clutch completion and hatching (see below for details).

In order to increase the costs of female parental investment, we induced blue tits to lay *ca* two extra eggs in experimental nests ($n = 34$ nests: 2013, $n = 13$; 2014, $n = 21$). Experiments on great tits (*Parus major*) show that females can be induced to lay an additional egg by removing the first two eggs on the days

each is laid [13,23,24]. Confamilial blue tits are also indeterminate egg layers [25], so by removing the first four eggs on the days each was laid, we expected to induce females to lay *ca* two (approx. 25%) more eggs. No blue tits in our experiment abandoned their nesting attempt when the first egg was removed. Removed eggs were placed under the nest in a padded plastic container 1.8 cm high and 5 cm in diameter, with a replaceable cardboard lid to prevent any moisture transfer from the nest. Our use of Woodcrete boxes with removable front doors allowed nests to be raised slightly without damage, and so keep removed eggs within the natural nest-box environment. After removal of the first four eggs, the female was allowed to lay the rest of her clutch, which she did without exception, before the removed eggs were then reinserted into the nest cup at the onset of incubation.

Without further intervention, however, experimental nests could be expected to have two more eggs than controls, as well as eggs deriving from under-nest conditions and from later in the laying sequence on average. We addressed these issues with two further procedures. First, we permanently removed (and froze for future work) the first egg laid from both experimental and control nests (controls received a dummy egg). Second, we then moved one egg from the experimental nests to replace the egg removed from control nests prior to incubation ($n = 16$ nests: 2013, $n = 7$; 2014, $n = 9$). (The number of experimental nests exceeded controls because of a concurrently running experiment—but eggs were removed in the same way from all 34 experimental nests.) Together, these further treatments ensured that experimental and control nests could be expected to have comparable clutch sizes at the onset of incubation (see results). Further, by selecting the translocated egg from experimental nests at random from those that were under the nest or those laid later in the clutch, we ensured that combined, control nests contained eggs that were exposed to under-nest conditions and derived from late in the laying sequence. We found no difference in the hatching success of control and experimental nests, nor differences in egg volumes (see results), and there is no compelling evidence in blue tits that clutches comprising eggs deriving from slightly later in laying sequences (1–2 eggs) differ in content from those laid earlier [26,27].

## (b) Prenatal treatment effects

To test the effects of the experiment on the number of eggs laid, we simply counted the total number of eggs laid by experimental and control females. To test for potential confounding differences between treatment groups, we also compared the average volume of eggs laid, number of eggs incubated, hatching success and hatching synchrony between experimental and control nests. The overall numbers of eggs laid and incubated were known with precision in all cases through repeated nest visits towards the end of laying and during early incubation. Control and experimental nests were visited a similar number of times; on average every 1–2 days from laying the first egg until the start of incubation (within first 10 days of laying: experimental mean: $9 \pm 1.5$ times; control mean: $8 \pm 2.3$ times; Welch's *t*-test ($n = 34,16$): $t_{20.80} = 1.50$, $p = 0.15$). In 2014, we calculated egg volumes from digital images of eggs ($n = 279$ eggs from nine control and 20 experimental clutches) (electronic supplementary material). All nests were checked daily for hatching from 11 days after the last egg was laid, to determine exact hatching date and the number of hatchlings (by counting the number of eggs that hatched successfully). Finally, hatching synchrony was estimated from the variance in nestling mass measured within 3 days after the start of hatching. This time-window was chosen because all viable eggs hatch within 3 days of each other in our population, but differences in nestling mass are still primarily determined by variation in hatching synchrony rather than provisioning rates

within this time frame ($n = 16$ experimental and eight control nests).

## (c) Postnatal treatment effects: provisioning behaviour and nestling mass

We used video recordings of parental feeding behaviour to determine female and male contributions to nestling provisioning (Sony HDR-CX220E Handycam® Camcorders; Shanghai, China). All videos were two hours long, but we excluded the first and last 10 min from analysis to minimize the effects of our presence on parental care behaviour. A single hour-long observation has been shown to be representative of individual provisioning rates in confamilial great tits [28]. Further, we can confirm that there is significantly repeatability in individual provisioning rates based on 100 min of provisioning data collected for other purposes on days 12 and 15 of the nestling period in 2015–2016 (repeatability analyses conducted using rpt-R controlling for brood age: $r = 0.46$, 95% confidence interval (CI) $= 0.29$–$0.61$, $p < 0.001$; $n = 194$ provisioning periods from 55 nesting pairs). In the present study, we analysed 85 h of video at the 50 nests. Videos were analysed blindly with respect to treatment group. Females and males were identifiable by their unique colour-band combinations (at least one member of the pair at each nest was colour-ringed). Blue tits are single-prey loaders in our population, meaning that parental birds only bring one prey item per visit to the nest-box. From each video, we extracted female and male provisioning events and the proportion of nest visits containing caterpillars. Prey items are generally small, about the size of a blue tit bill volume, but caterpillars have estimated volumes of 5–10 times larger, and have higher protein content than other prey items delivered (typically spiders and small adult arthropods) [29–31]. We therefore specifically tested whether the proportion of caterpillars delivered differed between control and experimental nests and fitted the proportion of caterpillars delivered as a covariate in models of provisioning rates. Feeding was recorded when broods were 9–17 days old (mean $= 13 \pm 1.4$ d) to ensure both parents were actively feeding at peak rates as females reduce brooding after the first-week post-hatching (fledging occurs from day 17 to 26 in our population (mean $= 21 \pm 1.2$ d)). Brood age at the time of video recording did not differ between experimental and control nests (experimental mean: $13 \pm 1.2$ d; control mean: $13 \pm 1.7$ d; Welch's *t*-test ($n = 34,16$): $t_{22.40} = -0.35$, $p = 0.73$). Nevertheless, brood age was fitted as a covariate in all models of parental provisioning rates (see below). Finally, all nestlings were weighed ($\pm 0.1$ g) on day 13–16 post hatching (mean $= 15 \pm 0.7$ d) in order to evaluate whether our measures of nestling provisioning behaviour captured meaningful variation in bi-parental contributions over the course of nestling dependence. No differences in the duration of the nestling period were found that may confound our estimates of nestling weight owing to differing development (experimental mean: $22 \pm 1.4$ d; control mean: $21 \pm 1.2$ d; Welch's *t*-test ($n = 34,16$): $t_{25.28} = 0.68$, $p = 0.51$).

## (d) Statistical analysis

Statistics were performed in *R* v. 3.5.0 [32]. Normal response variables and models generating normally distributed residuals were analysed using Welch's *t*-tests or linear models (LMs) in the 'stats' package [32]. Non-normal response terms and/or those resulting in non-normal distributions of residuals were analysed using Mann–Whitney–Wilcoxon tests ('stats' package). Mixed models were run in the nlme package [33] when response variables contained non-independent data. All models underwent checks for overdispersion and heteroscedasticity of residuals [34]. We used *t*-tests/Mann–Whitney–Wilcoxon tests to verify that our experiment indeed changed the number of

eggs laid, but no other features of clutches or broods were impacted. By contrast, LMs were used to analyse the impact of the experiment on parental provisioning rates and nestling mass while accounting for covariates known to typically influence the response variable (see below). The significance of terms in models was evaluated using changes in deviance between full models and models excluding each term using the ANOVA function in R (significance set at $\alpha < 0.05$ [34]; with terms that failed to contribute significant explanatory power removed from the final model. However, treatment was retained in all models as this was our primary variable of interest.

First, we used Welch's $t$-tests or Mann–Whitney–Wilcoxon tests, depending on data normality, to investigate differences between experimental nests and controls in the total number of eggs laid, as well as in the average volume of eggs laid per clutch (2014 only), the number of eggs incubated, hatching success, hatching synchrony and proportion of caterpillars delivered. Hatching synchrony, measured as variance in brood mass, was normalized using Tukey's Ladder of Powers, which determines the power transformation that most closely fits the data to a normal distribution [35]. Second, we investigated the effect of the experiment on female and male provisioning rates using two LMs with Gaussian error structure and identity link function. We analysed the sexes separately to determine the independent effects of treatment on each. In these analyses, the following potential confounding covariates were included: brood age and brood size and their polynomial (quadratic effects), lay date, altitude, year and the proportion of caterpillars delivered which could confound estimates of provisioning rates and nestling mass (see the electronic supplementary material, tables S1A,B for further details). Both controls and experimental nests had comparable proportions of first-year individuals (56% and 74% yearlings in control and experimental nests, respectively). The inclusion of parental age in the models did not alter our conclusions but reduced the sample size since not all parents were captured; so we excluded this term from the models (electronic supplementary material). Finally, we tested the effect of treatment on total brood provisioning rates and nestling mass within broods. In the case of total provisioning rate, we used an LM with normal errors and identity link function and included the same potential confounders as for the sex-specific analyses outlined above. In the nestling mass analysis, we fitted individual nestling mass in a linear mixed model (LMM) with maximum-likelihood (ML) estimation, with brood age and size, variance in hatching mass, lay date, altitude and year as potential confounding variables and nest identity as a random factor to account for non-independence of the masses of nestlings from the same nest.

## 3. Results

### (a) Prenatal treatment effects

Control females laid clutches of 8.4 eggs on average (s.d. = 0.96, range = 7–10). Removing the first four eggs on the day each was laid resulted in experimental females laying two extra eggs on average compared with controls, an increase in clutch size of approximately 25% (Welch's $t$-test ($n = 34,16$): $t_{34.88} = 6.23$, $p < 0.001$; figure 1a). Subsequent removal of one egg from all nests and translocation of another from experimental to control nests at the onset of incubation ensured that both sets of females incubated clutches of comparable size (mean ± s.d. = 8.3 ± 1.1 (control) versus 8.4 ± 1.2 (experimental); Mann–Whitney–Wilcoxon test: W ($n = 34,16$) = 276.5, $p = 0.93$). Further, there was no difference between experimental and control clutches in terms of the volume of eggs laid (mean ± s.d. = 1.1 cm$^3$ ± 0.05 (control) versus 1.1 ± 0.1 (experimental); $t$-test ($n = 20,9$): $t_{26.99} = -1.16$, $p = 0.26$; 2014 only) or

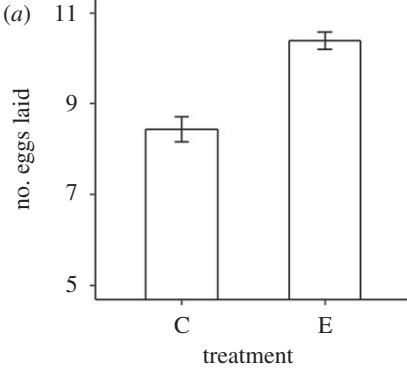

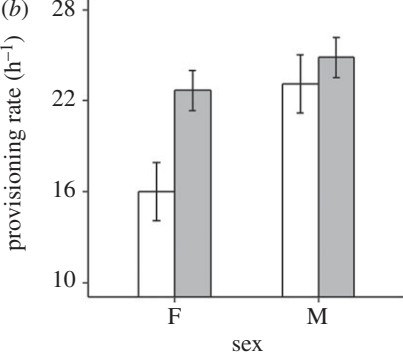

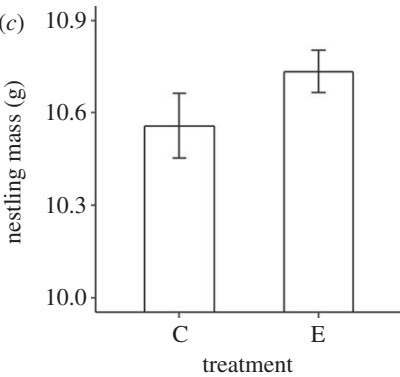

**Figure 1.** Treatment effects. (a) Experimental (E) females laid approximately 25% more eggs than controls (C). (b) Experimental (grey bars) females (F) fed nestlings approximately 43% more frequently than controls (white bars), while experimental males (M) did so approximately 9% more frequently. (c) Increased feeding in experimental pairs resulted in nestlings being 6% heavier than controls. Figures show predicted means (± s.e.) from linear models (mixed model in the case of mean nestling mass), after controlling for significant effects of brood size (female analysis); brood size, brood age, lay date and year (male analysis); and brood age, lay date and altitude (nestling mass analysis).

hatching success (mean ± s.d. = 0.9 ± 0.12 (control) versus 0.9 ± 0.15 (experimental); Mann–Whitney–Wilcoxon test: W ($n = 34,16$) = 274.5, $p = 0.97$). Finally, there was no difference between treatment groups in either hatching synchrony (measured as variance in nestling mass on day 3: mean ± s.d. = 0.8 ± 0.5 (control) versus 1.1 ± 1.2 (experimental), $t$-test ($n = 24,11$): $t_{29.46} = 0.94$, $p = 0.35$) or brood size on the day video recording occurred (mean ± s.d. = 7 ± 1.8 (control) versus 7 ± 1.9 (experimental); $t$-test ($n = 34,16$): $t_{30.84} = 0.33$, $p = 0.74$). Given that the ages of broods were also comparable between treatment groups on the days videos were recorded (see Material and methods), the key difference between experimental and control females appears to be in the number of eggs laid.

## (b) Provisioning behaviour of females and males

During peak provisioning, control females fed broods at an average rate of 16 prey items h$^{-1}$ (s.d. = 8, range = 3–30), while control males did so at an average rate of 23 prey items h$^{-1}$ (s.d. = 10, range = 4–42). Contrary to the prediction that current–future trade-offs can operate between distinct phases *within* the same breeding event [14], females in experimental nests delivered prey to their broods 43% more frequently than those from control nests (LM: $F_{1,46} = 11.11$, $p \leq 0.002$; figure 1b; electronic supplementary material, table S1A). Given that females in experimental nests showed a marked elevation in their provisioning rates, traditional models [5–7] would predict that males in such nests should partially reduce their own contribution, but this was not apparent. Instead, males showed a non-significant tendency to respond to the increased provisioning rates of their partners in experimental nests by increasing (by 9%), not decreasing, their own rates of provisioning (LM: $F_{1,43} = 2.86$, $p = 0.098$; figure 1b; electronic supplementary material, table S1B).

Any treatment effects on provisioning rates cannot be confounded by the number of prey items delivered per visit, for blue tits in our population are single-prey loaders. Further, on the whole, caterpillars formed a relatively small proportion of all prey delivered (approx. 10% total in the two years), and there were no differences in the proportion of nest visits containing caterpillars between the two treatment groups for either females (median (inter-quartile range (IQR)) = 0.04 (0–0.1) (control) versus 0.03 (0–0.08) (experimental); Mann–Whitney–Wilcoxon test: W ($n = 34,16$) = 257.5, $p = 0.76$) or males (median (IQR) = 0.09 (0.03–0.17) (control) versus 0.08 (0.04–0.13) (experimental); Mann–Whitney–Wilcoxon test: W ($n = 34,16$) = 270, $p = 0.98$). Finally, adding the proportion of caterpillars delivered into models of treatment effects on provisioning rates failed to have a significant impact on the results (for females: $p = 0.15$; for males: $p = 0.63$; electronic supplementary material, tables S1A,B). As a consequence, the elevated rates of prey delivery by females and the non-significant trend in the same direction for male partners in experimental nests cannot easily be explained by tendencies to deliver relatively small prey, at least as measured by the rate at which caterpillars are delivered.

## (c) Treatment effects on brood provisioning rates and nestling mass

The average total rate at which control broods were provisioned was 39 prey items h$^{-1}$ (s.d. = 15, range = 8–69). Given that both females and males elevated their provisioning rates in experimental nests, it is unsurprising that overall provisioning rates were also significantly greater in experimental nests (LM ($n = 34,16$): $F_{1,43} = 15.31$, $p < 0.001$; electronic supplementary material, table S2A). A more important question is whether or not the 22% average increase in the rate at which experimental broods were provisioned relative to controls translates into differences in nestling mass, which would elucidate whether the magnitude increase in provisioning rates could be functionally meaningful. The average mass of nestlings in control broods was 10.8 g ($\pm 1.1$ s.d.), with among-brood variation explained in part by variation in lay date, brood age and altitude, but not brood size (electronic supplementary material, table S2B). After controlling for these parameters, nestlings in experimental broods were

significantly heavier than those from control broods (LMM ($n = 263,133$ nestlings): $\chi_1^2 = 3.90$, $p = 0.048$; figure 1c). Although the magnitude of the increase was not particularly marked, the 6% increase is in line with expectation based on an estimated 20% conversion of food intake into mass (i.e. $0.2 \times 0.2 = 0.4$) (see discussion). The key point is that the increased provisioning rates documented in response to the experiment were, at least in part, reflected in increased nestling mass.

## 4. Discussion

Experimental females that were induced to lay approximately two more eggs than control females provisioned their offspring 43% more often than control females. Contrary to the classic expectation of partial compensation, male partners at experimental nests did not appear to reduce their provisioning rates and indeed showed a non-significant tendency to have greater provisioning rates (9% more) than controls. As a consequence, offspring in experimental nests were provisioned at a significantly higher overall rate and were heavier than those in control nests. The 6% increase in nestling mass is roughly what would be expected from a 22% increase in feeding rate given that only a small proportion of food delivered is typically converted to chick growth (approx. 20%; [29,36]). Together, these findings have two broad implications. First, given that prenatal and postnatal investment appear to be positively associated in female blue tits suggests that expected current–future life-history trade-offs are not operating between discrete phases within reproductive events in this system. Instead, this positive association suggests that increased high prenatal investment should be associated with reduced future reproductive value and that a common mechanism links pre- and postnatal female investment. Second, these results suggest that variation in prenatal investment by the female shapes her 'assessment' of current brood value and that her subsequent positive adjustments in postnatal provisioning rates are used by her male partner to guide his own contributions.

Our results are not easily explained by known confounders of parental feeding rates. First, experimental and control nests did not differ in lay date or altitude and, given that they were additionally 'paired' by distance, are unlikely to have occupied territories of differing quality. Nor did they differ in the number of eggs incubated, which might impact incubation costs [19], or brood size, which can impact brood hunger and begging [37,38]—because clutch sizes were made comparable between experimental and control nests before the onset of incubation. Second, if females perceived egg removal as predation, we would either expect experimental females to reduce their provisioning rate (i.e. save resources for the future; [39–41]) or any increases in provisioning rates to hasten the developmental rate of their brood [42,43]. On the contrary, not only did we find that experimental females increased their provisioning rates, but that experimental and control broods fledged at a comparable age (see material and methods). Third, although experimental nests contained eggs that were laid *ca* one egg later in the laying sequence on average than control nests, there is little evidence to suggest that such slight differences would be confounding, particularly over a brood averaging eight nestlings. While there is a weak increase in lipid content

across the laying sequence of blue tits in other populations [26], both egg volume and hatching synchrony (i.e. variance in size at hatch), which can impact brood demand [44,45], were comparable between experimental and control nests in our study. In addition, a previous study in blue tits failed to detect a consistent change in testosterone levels deposited in eggs across the laying sequence, which can impact nestling begging intensity [27]. Finally, we found no obvious confounding effects of prey load size, because the proportion of caterpillars delivered did not differ between control and experimental nests and treatment effects on provisioning rates controlled for the proportion of caterpillars delivered.

Life-history theory can make contrasting predictions about the relationship between contributions to pre- versus postnatal investment within the same breeding event depending on the timescale over which expected current–future trade-offs operate [3,14,18,19,46]. Although previous studies have not disentangled the relationship between pre- and postnatal investment directly, the few experimental studies of relevance suggest that trade-offs can operate across distinct phases of the same reproductive event. For example, common terns (*Sterna hirundo*) only successfully rear an extra nestling if they do not have to invest in its production or incubation [19]. Similarly, Monaghan *et al.* [14] showed that inducing lesser black-backed gulls (*Larus fuscus*) to rear an extra egg resulted in fewer fledglings produced and females having reduced body condition. By contrast, our study provides, to our knowledge, the first test of the direct association between prenatal investment and contributions to postnatal care and shows that investment across the two phases of a breeding event positively covary: female blue tits induced to lay two extra eggs (i.e. increased prenatal investment) also increased their postnatal investment, (i.e. 43% more provisioning than controls) even though they were not provisioning more young. Such a positive association is expected if current–future life-history trade-offs operate more strongly at the level of breeding events (than phases within), because high prenatal investment is expected to reduce the future reproductive value, and so select for higher overall levels of investment into current offspring [3,18]. Direct contrasts of adult survival between control and experimental nests would help confirm this trade-off, but low inter-annual return rates of adults in our populations (approx. 21%, presumably explained by the expansive and contiguous network of suitable habitat in the French Pyrenees) make such contrasts impossible in this current study.

Why pre- and postnatal investment should apparently trade-off in terns and gulls, but covary positively in blue tits is not currently known. One possibility is that the impacts of prenatal investment on the relative importance of current versus residual reproductive value are a key factor in driving within versus across breeding attempt trade-offs [3,18]. Indeed, because gulls and terns have smaller clutch sizes and are longer-lived than tits, their proportional fitness returns from a current breeding attempt will be, on average, less relative to the fitness returns from future breeding attempts. As a consequence, longer-lived species with low fecundity might therefore be under greater selection to compensate for high prenatal costs by reducing postnatal investment to ensure survival to, and investment in, future reproductive events. By contrast, such compensation strategies will have a reduced impact on lifetime fitness of more

fecund, shorter-lived species, because fitness returns from current events are proportionally more important for lifetime fitness. Further manipulative studies are required to investigate actual impacts on survival, the timescale over which current–future trade-offs operate and the role of life-history traits and residual reproductive value in explaining the inevitable variation.

The ultimate explanation for variation in the timescale over which current–future trade-offs operate notwithstanding, previous studies on terns and gulls and ours here, suggest that mechanisms link pre- and postnatal investment. When trade-offs within a breeding event exist, such a mechanism might simply be resource or energy limitation experienced by the parent [14,19]. However, such a mechanism is unlikely to explain an experimentally induced positive relationship as was found here. Instead, our positive association suggests that hormonal mechanisms might link prenatal fecundity with postnatal investment. One such candidate hormone is prolactin, which is known to be associated with both offspring production and parental care [47,48], although a direct link between prolactin levels and quantitative changes in offspring production and parental care has yet to be shown [49]. Alternatively, a longer production period (or greater number) of offspring should lead to higher levels of gonadotropin-releasing hormone [47] which, in experimental tests with dark-eyed juncos (*Junco hyemalis*), leads to an increase in testosterone and higher feeding rates [50]. Either way, a hormonal mechanism that links pre- and postnatal care could be adaptive if the number of offspring produced sufficiently predicts the number requiring food postnatally, which it does in blue tits with low levels of hatching failure and brood reduction ([51], this study). Identifying which hormonal systems might be responsible for linking pre- and postnatal investment, and whether such a link is restricted to species with predictable associations between prenatal investment and postnatal demand for food remains to be investigated.

Traditional bi-parental care theory would predict that experimental male partners reduce their contributions to offspring provisioning to partially compensate the 43% average increase shown by their female partners [5,7]. Indeed, partial compensation is the most frequent response [8] and has been observed in blue and great tits ([52,53], but see [54]). However, we found little firm evidence for partial compensation, with males at experimental nests showing a non-significant tendency for greater (9%) provisioning rates relative to control males. This result suggests that males are either unresponsive to female changes or partially match them. Non-compensatory or partial matched responses are predicted under two circumstances. First, each is predicted if females are more informed as to the value of a current brood than their male partner, and the value of the current brood to the female is signalled in her provisioning behaviour [10]. Our findings in blue tits are consistent with this hypothesis: (i) the positive link between pre- and postnatal investment by females found in this study at least suggests that brood value might be linked with prenatal investment and be reflected in female provisioning rates; and (ii) blue tits are socially monogamous in our population and have relatively limited cuckoldry in others [55,56], leading the two sexes to have aligned values of the current brood [4]. Second, no compensation is expected when the variation in brood values and parental state are comparable, whereas matched responses are expected when variation in brood

value is greater [10]. Further studies are required to test the relative variation of brood value and parental state, but in this short-lived species, males, like females, might be expected to favour maintenance of high investment in broods of relatively high value [21]. Thus, in blue tits, males might not only benefit from using cues directly from the brood to guide their optimal provisioning investment but refine this adaptively based on the provisioning rate of their partner [54,57].

In conclusion, our results generate three hypotheses with important implications for our understanding of the eco-evolutionary drivers of life-history variation, patterns of parental care and/or the maintenance of bi-parental care. First, the degree to which current–future life-history trade-offs operate within versus across breeding events might be expected to vary as a function of the relative value between current and future breeding events for lifetime fitness. This hypothesis not only has important implications for our understanding of patterns of parental care but also in understanding variation in patterns of offspring quality–quantity trade-offs that are often hard to demonstrate [58,59]. Specifically, such trade-offs will be diluted when current–future trade-offs operate primarily across attempts. Second, an underlying hormonal mechanism might adaptively link pre- and postnatal investment in species with variable fecundity and wherein fecundity in a current event is a good predictor of postnatal demand in the same attempt. Such a mechanism, if it exists, would have important implications for understanding among-female variation in contributions to postnatal care as well as the magnitude of plastic responses to environmental variation [60]. Finally, where pre- and postnatal investment positively covaries in females, male partners in bi-parental systems might benefit by using the postnatal investment of their partner positively, not only to gauge current hunger levels of the brood, but also its potential relative fitness value, with implications for explaining exceptions to partial compensation response rules in bi-parental care systems.

Ethics. Work was conducted under animal care permits to A.S.C. from the French bird ringing office (CRBPO; no. 13619; PP576), the state of Ariège animal experimentation review (Préfecture de l'Ariège, Protection des Populations, no. A09-4) and the Région Midi-Pyrénées (DIREN, no. 2012-07).

Data accessibility. The data supporting the results available as part of the electronic supplementary material.

Authors' contributions. A.C.B., A.S.C. and A.F.R. conceived the study, all authors collected data, E.S. created and validated the R script to analyse egg volume, A.C.B. conducted the analyses, and A.C.B., A.S.C. and A.F.R. wrote the paper. All authors gave final approval for publication and agree to be held accountable for the work performed therein.

Competing interests. We have no competing interests.

Funding. This work was funded by fellowship grants from the Région Midi-Pyrénées (A.C.B., A.S.C., A.F.R.); the Centre National de la Research Scientifique (A.C.B., A.S.C.); a Natural Environment Research Council to the University of Exeter, as well as research grants from the Agence National pour la Recherche (ANR-JCJC 'Net-Select') and the Human Frontiers Science Partnership (RGP0006/2015 'WildCog') to A.S.C. This work is part of the Laboratoire d'Excellence (LABEX) entitled TULIP (ANR-10-LABX-41).

Acknowledgements. We thank the landowners who granted permission to work on private land, J.-P. Molinier and the Office National des Fôrets for access to public land, the many volunteers that assisted with fieldwork, video and photo processing, in particular, C. McNicol and B. Harris, and finally P. Heeb, B. Lyon, N. Royle, J. Savage, S. Sharp, B. Tschirren and two anonymous reviewers for helpful comments on a previous version of the manuscript.

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
