## [Reviewer comments · Proceedings of the Royal Society B: Biological Sciences]

Review History

RSPB-2019-0395.R0 (Original submission)

Review form: Reviewer 1

Recommendation

Major revision is needed (please make suggestions in comments)

Scientific importance: Is the manuscript an original and important contribution to its field?

Excellent

General interest: Is the paper of sufficient general interest?

Excellent

Quality of the paper: Is the overall quality of the paper suitable?

Acceptable

Is the length of the paper justified?

Yes

Should the paper be seen by a specialist statistical reviewer?

No

Do you have any concerns about statistical analyses in this paper? If so, please specify them explicitly in your report.

No

It is a condition of publication that authors make their supporting data, code and materials available - either as supplementary material or hosted in an external repository. Please rate, if applicable, the supporting data on the following criteria.

Is it accessible?

N/A

Is it clear?

N/A

Is it adequate?

N/A

Do you have any ethical concerns with this paper?

No

Comments to the Author

General

The main finding was very interesting and also surprising. This is because blue tits (BT) are short-lived, and one would expect from life history theory that also the females in the control group would invest heavily in rearing offspring when they have succeeded to find a suitable nest cavity (which may be in short supply) and found a mate. Therefore, I am a bit skeptical to the conclusion. This is for three reasons.

(1) Sample size was OK for experimental females ($n = 34$) but rather small for controls ($n = 16$).

(2) The conclusion is based on the difference in provisioning rate, namely that it was 43% higher for experimental females than for controls.

(i) Previous studies have shown that provisioning rate in BT shows considerable variation, and thus the result may be sensitive to the small sample for controls. Provisioning rate was based on only 2h of filming once, and from these data the authors excluded the first and last 10 min of the video (to avoid effects of human visiting), leaving a relatively low amount of filming per female (100 min). Birds may forage for a while e.g. in one tree, but after some time change to a different tree of a different species. Thus, provisioning of prey items may be consistent over a time period but then change. With low sample of birds, this may cause a problem.

(ii) With such a great difference in provisioning between female groups, and that males of experimental females had a 9% higher provisioning rate than males of control females, one would expect a strong difference in nestling body mass, but the effect was close to non-significant ($p = 0.048$, $l. 335$), which is very hard to explain.

(ii) From the videos, the total number of prey items, and the proportion of green caterpillars, were recorded but not prey size. I suggest that the authors use the videos to analyse (a "blind" person) prey size to study whether or not the items were smaller in case of experimental females (which may explain the low effect on nestling mass). A way may be to compare prey size relative to beak size. Many species of caterpillars were probably involved, and different stages of development and therefore sizes.

(3) The authors conclude that their finding is consistent with the hypothesis of a trade-off between breeding events (positive prenatal-postnatal investment association within a season). However, their study provides no evidence for predicted costs to the experimental females of increasing their provisioning rate, like a drop in female body mass across the nesting period, reduced subsequent survival or a reduction in reproductive success in the following year. The authors seem to have ringed all the birds, and thus it may be possible to compare female survival rate between the two groups.

(4) It is unknown how the removal of four eggs from the experimental females was perceived, i. e. the effect observed of provisioning may have been an "artefact" of the experimental treatment. For instance, the egg removal may have been perceived as a sign of high nest predation risk, affecting subsequent female behavior by investing more heavily when she has the opportunity (a life history argument), or to increase nestling growth rate so that the offspring would advance the time of fledging and escape nest predation. This is of course poor speculation but such possible effects of treatment may also be discussed.

Specific points

(5) l. 56. You mention "maternal production costs". This seems misleading because you do not seem to have recorded costs to females that are usually recorded in life history studies, namely loss of female body mass, delayed time of molting, reduced subsequent survival rate and reproduction.

(6) l. 145. You say that it was a colour-ringer population. However, then it may be possible to measure survival rate and future reproductive success. On the other hand, you say later (l. 152) that (only) two females were found in both seasons - why so few? Annual survival rate in adult BT is usually about 40%.

(7) According to life history theory, age of the bird may be important for its investment. Was there any effect of female age - was the proportion of yearlings the same in the two groups?

(8) p. 7. Sample sizes were relatively small and so you may mention how many in each year.

(9). Each evening during egg laying the female may warm the eggs for a few minutes. This may improve hatching success (fighting microbes). How was the hatching success of the "under-nest-eggs"?

(10) p. 10. You may already here in methods say that the BT is a single prey loader. You say it on l. 311 under the section of Statistical analysis.

(11) Results. I suggest you add sample size for all variables/tests.

(12) l. 299-312 + 328-321. I suggest you present only the results here; now it is a mix of results and discussion, causes overlap/repetition with the discussion.

(13) l. 313. I suggest you give the mean percentages for the amounts of green caterpillars here, so that the readers can tell what kind of prey was available (a "good" or "bad" season). The proportion may differ substantially among years.

(14) l. 436. You say that BTs are socially monogamous, but this is not the case in every population.

(15) Figure legends. A figure should be possible to understand from the legend. Thus, I suggest

that you say a bit more of what experimental females meant.

Review form: Reviewer 2

Recommendation

Accept with minor revision (please list in comments)

Scientific importance: Is the manuscript an original and important contribution to its field?

Good

General interest: Is the paper of sufficient general interest?

Good

Quality of the paper: Is the overall quality of the paper suitable?

Excellent

Is the length of the paper justified?

Yes

Should the paper be seen by a specialist statistical reviewer?

No

Do you have any concerns about statistical analyses in this paper? If so, please specify them explicitly in your report.

Yes

It is a condition of publication that authors make their supporting data, code and materials available - either as supplementary material or hosted in an external repository. Please rate, if applicable, the supporting data on the following criteria.

Is it accessible?

N/A

Is it clear?

N/A

Is it adequate?

N/A

Do you have any ethical concerns with this paper?

No

Comments to the Author

See attached report (Appendix A).

Decision letter (RSPB-2019-0395.R0)

02-Apr-2019

Dear Dr Bruendl:

I am writing to inform you that your manuscript RSPB-2019-0395 entitled "Experimentally induced increases in fecundity lead to greater nestling care in blue tits" has, in its current form, been rejected for publication in Proceedings B.

This action has been taken on the advice of referees, who have recommended that substantial revisions are necessary. With this in mind we would be happy to consider a resubmission, provided the comments of the referees are fully addressed. However please note that this is not a provisional acceptance.

In your revision process, please take a second look at how open your science is; our policy is that all data involved with the study should be made openly accessible-- see: <https://royalsociety.org/journals/ethics-policies/data-sharing-mining/> Insufficient sharing of data can delay or even cause rejection of a paper.

Sincerely,
Professor John R. Hutchinson, Editor
Proceedings B
<mailto:proceedingsb@royalsociety.org>

Associate Editor
Board Member: 1
Comments to Author:

We have hereby received two expert reviews for this manuscript investigating hypotheses on the balance in parental care between individuals and within individuals. Both reviewers liked the experimental approach, found the results interesting, and the manuscript generally well written.

However, they both also made substantial comments, that should improve the quality of the manuscript and the arguments. Reviewer 1 expressed reserve at the small sample size for the control females, and it would be good if the authors could think of a way to alleviate this, e.g. through reasoning, or statistically. The suggested additional analyses from the videos would go some way too, to make the results more robust and should additional video footage exist, it would be worthwhile to expand the number of hours observed. Additionally, both reviewers made several suggestions on how to improve the framing and interpretation of the results, that will improve the impact of the manuscript, and particularly the suggestion that the manipulation might have had unintended effects on perception of predation risk seems important - is there any relevant information in the literature to support or discount this suggestion. Both reviewers have made thoughtful and constructive suggestions that should improve the quality of the manuscript.

Reviewer(s)' Comments to Author:

Referee: 1

Comments to the Author(s)

General

The main finding was very interesting and also surprising. This is because blue tits (BT) are short-lived, and one would expect from life history theory that also the females in the control group would invest heavily in rearing offspring when they have succeeded to find a suitable nest cavity (which may be in short supply) and found a mate. Therefore, I am a bit skeptical to the conclusion. This is for three reasons.

(1) Sample size was OK for experimental females ($n = 34$) but rather small for controls ($n = 16$).

(2) The conclusion is based on the difference in provisioning rate, namely that it was 43% higher for experimental females than for controls.

(i) Previous studies have shown that provisioning rate in BT shows considerable variation, and thus the result may be sensitive to the small sample for controls. Provisioning rate was based on only 2h of filming once, and from these data the authors excluded the first and last 10 min of the video (to avoid effects of human visiting), leaving a relatively low amount of filming per female (100 min). Birds may forage for a while e.g. in one tree, but after some time change to a different tree of a different species. Thus, provisioning of prey items may be consistent over a time period but then change. With low sample of birds, this may cause a problem.

(ii) With such a great difference in provisioning between female groups, and that males of experimental females had a 9% higher provisioning rate than males of control females, one would expect a strong difference in nestling body mass, but the effect was close to non-significant ($p = 0.048$, $l. 335$), which is very hard to explain.

(ii) From the videos, the total number of prey items, and the proportion of green caterpillars, were recorded but not prey size. I suggest that the authors use the videos to analyse (a "blind" person) prey size to study whether or not the items were smaller in case of experimental females (which may explain the low effect on nestling mass). A way may be to compare prey size relative to beak size. Many species of caterpillars were probably involved, and different stages of development and therefore sizes.

(3) The authors conclude that their finding is consistent with the hypothesis of a trade-off between breeding events (positive prenatal-postnatal investment association within a season). However, their study provides no evidence for predicted costs to the experimental females of increasing their provisioning rate, like a drop in female body mass across the nesting period, reduced subsequent survival or a reduction in reproductive success in the following year. The

authors seem to have ringed all the birds, and thus it may be possible to compare female survival rate between the two groups.

(4) It is unknown how the removal of four eggs from the experimental females was perceived, i. e. the effect observed of provisioning may have been an “artefact” of the experimental treatment. For instance, the egg removal may have been perceived as a sign of high nest predation risk, affecting subsequent female behavior by investing more heavily when she has the opportunity (a life history argument), or to increase nestling growth rate so that the offspring would advance the time of fledging and escape nest predation. This is of course poor speculation but such possible effects of treatment may also be discussed.

Specific points

(5) l. 56. You mention “maternal production costs”. This seems misleading because you do not seem to have recorded costs to females that are usually recorded in life history studies, namely loss of female body mass, delayed time of molting, reduced subsequent survival rate and reproduction.

(6) l. 145. You say that it was a colour-ringer population. However, then it may be possible to measure survival rate and future reproductive success. On the other hand, you say later (l. 152) that (only) two females were found in both seasons – why so few? Annual survival rate in adult BT is usually about 40%.

(7) According to life history theory, age of the bird may be important for its investment. Was there any effect of female age – was the proportion of yearlings the same in the two groups?

(8) p. 7. Sample sizes were relatively small and so you may mention how many in each year.

(9). Each evening during egg laying the female may warm the eggs for a few minutes. This may improve hatching success (fighting microbes). How was the hatching success of the “under-nest-eggs”?

(10) p. 10. You may already here in methods say that the BT is a single prey loader. You say it on l. 311 under the section of Statistical analysis.

(11) Results. I suggest you add sample size for all variables/tests.

(12) l. 299-312 + 328-321. I suggest you present only the results here; now it is a mix of results and discussion, causes overlap/repetition with the discussion.

(13) l. 313. I suggest you give the mean percentages for the amounts of green caterpillars here, so that the readers can tell what kind of prey was available (a “good” or “bad” season). The proportion may differ substantially among years.

(14) l. 436. You say that BTs are socially monogamous, but this is not the case in every population.

(15) Figure legends. A figure should be possible to understand from the legend. Thus, I suggest that you say a bit more of what experimental females meant.

Referee: 2

Comments to the Author(s)

See attached report.

Author's Response to Decision Letter for (RSPB-2019-0395.R0)

See Appendix B.

RSPB-2019-1013.R0

Review form: Reviewer 1 (Tore Slagsvold)

Recommendation

Accept as is

Scientific importance: Is the manuscript an original and important contribution to its field?

Excellent

General interest: Is the paper of sufficient general interest?

Excellent

Quality of the paper: Is the overall quality of the paper suitable?

Good

Is the length of the paper justified?

Yes

Should the paper be seen by a specialist statistical reviewer?

No

Do you have any concerns about statistical analyses in this paper? If so, please specify them explicitly in your report.

No

It is a condition of publication that authors make their supporting data, code and materials available - either as supplementary material or hosted in an external repository. Please rate, if applicable, the supporting data on the following criteria.

Is it accessible?

N/A

Is it clear?

Yes

Is it adequate?

Yes

Do you have any ethical concerns with this paper?

No

Comments to the Author

I think the authors have done a good job. Some important data/information is still missing (like female body condition and survival rate) but I understand that this is hard to get.

Decision letter (RSPB-2019-1013.R0)

20-May-2019

Dear Dr Bruendl

I am pleased to inform you that your manuscript RSPB-2019-1013 entitled "Experimentally induced increases in fecundity lead to greater nestling care in blue tits" has been accepted for publication in Proceedings B.

The referee(s) have recommended publication, but also suggest some minor revisions to your manuscript. Therefore, I invite you to respond to the referee(s)' comments and revise your manuscript. Because the schedule for publication is very tight, it is a condition of publication that you submit the revised version of your manuscript within 7 days. If you do not think you will be able to meet this date please let us know.

Once again, thank you for submitting your manuscript to Proceedings B and I look forward to receiving your revision. If you have any questions at all, please do not hesitate to get in touch. And again, congratulations!

Sincerely,

Professor John R. Hutchinson, Editor

Proceedings B

Associate Editor

Board Member

Comments to Author:

The authors did a good job on the revisions, as the reviewer testifies also. In regards to the figures, I feel that perhaps a different choice of figures to present, or one combination figure of multiple panels, would represent the results most clearly. This would include the effects on # eggs (previous figure 1), the male and female feeding rate (omitting the total, which is the sum of male and female) and the effect on nestling body mass. Such a figure would represent the whole story. Given that these are simple bar charts, a composite figure, or separate figures, should not take up too much space. For example, the axis labels could be C and E, and the bars narrower.

Reviewer(s)' Comments to Author:

Referee: 1

Comments to the Author(s).

I think the authors have done a good job. Some important data/information is still missing (like female body condition and survival rate) but I understand that this is hard to get.

Author's Response to Decision Letter for (RSPB-2019-1013.R0)

See Appendix B.

Decision letter (RSPB-2019-1013.R1)

03-Jun-2019

Dear Dr Bruendl

I am pleased to inform you that your manuscript entitled "Experimentally induced increases in fecundity lead to greater nestling care in blue tits" has been accepted for publication in Proceedings B.

Open Access

Paper charges

Sincerely,

Editor, Proceedings B
mailto: proceedingsb@royalsociety.org

Appendix A

Summary

This study investigates the trade-off between prenatal and postnatal investment in offspring by females, and the consequences for postnatal male investment. The authors hypothesize that increased pre-natal investment by the female will result in a reduction in postnatal investment in female care, and an increase in male care to compensate. To test this hypothesis, blue tit females were experimentally induced to lay additional eggs, but to raise similar numbers of chicks as control groups, and the volume of the eggs, the male and female provisioning rates, and the weight of the nestlings were measured. Contrary to predictions, females that were induced to lay more eggs provided more postnatal care, and males did not compensate for the change in female investment. The offspring of experimental groups were also heavier, suggesting that the increase in total investment was beneficial in terms of offspring fitness. The authors discuss their results in the context of life history theory and the evolutionary maintenance of biparental care.

General Comments

This study provides a clear test of how maternal egg production costs can influence future investment in the same brood, and the consequences for partner provisioning. The methods and experimental design clearly test the hypothesis, and the manuscript as a whole is very well written. The authors present a novel test of how trade-offs between current and future reproduction operate at the scale of a breeding event, and I believe it will be of interest to the broad readership of PRSB. My comments on this manuscript are focused on some points that I believe need clarification in the methods and results, and I provide some suggestions on how to better tie together the introduction and discussion. In general, I feel that this manuscript is an important contribution to our understanding of the scale on which life history trade-offs operate, and how that influences patterns of parental care.

Specific Comments

105-111: Here you provide a nice explanation for two contrasting predictions. I suggest reiterating this at the end of the introduction along with a statement of your other predictions

118-122: Given that your results did not really support either compensation or matching, I suggest rewording this section. In the model you cite by Johnstone & Hinde (2006) there are contexts that result in no, partial, or full compensation, as well as matching. If you state that all are potential outcomes it will help set up the discussion of your results, particularly of no compensation or partial matching by the male.

137: Here I suggest providing an explicit statement of your hypotheses and predictions, which are not that clear from your introduction, but you explain in the discussion (e.g. lines 373-375 and lines 431-434). For example, first you ask how pre and postnatal investment will covary and have two contrasting predictions, and second, you ask how the male responds to changes in partner effort and predict alternate outcomes based on

the amount of information a male has on the value of the brood. I think being explicit here will help improve the flow between the introduction and the discussion and help justify your explanation of why your results did or did not match your predictions.

150: I suggest also including a summary of the sample sizes: the total number of nests used, the number per treatment, and the number per year, either here or at the beginning of your experimental design section.

152-154: Can you clarify that all other females were found only once, in either 2013 or 2014?

207: By egg size do you mean the volume? I would state this specifically, since size could mean mass or surface area

211-212: Can you state the average number of times nests were visited?

217: There is a typo, this should be “before **and** three days after the start of hatching”

258-269: It is not clear to me why total number of eggs laid, average volume of eggs laid per clutch, number of eggs incubated, number of eggs hatching and hatching synchrony were analyzed in a t-test while lay date, brood age and size, altitude, year and the proportion of caterpillars delivered were covariates in the linear model. I suggest either explaining your reasoning (e.g. to confirm that X variables did not differ between the two treatments we used a t-test, while to control for the potential confounding effects of Y variables on parental provisioning we used a linear model) or conducting some model selection procedure (e.g. stepwise regression) to determine which covariates to include.

312-319: Here you use a t-test to compare proportion of caterpillars delivered in addition to including it as a covariate in the linear model. But in the methods (see comment on lines 258-269) you only have it as a covariate. I believe this can be easily cleared up by more explicitly stating in the methods which variables you used in each test of your hypotheses and why.

342: There is a typo, this should be “male partners at experimental **nests** did not appear to reduce their provisioning rates”

427-429: Have there been tests of the compensation hypothesis in blue tits or a related species? If so, I suggest mentioning those studies, which could help explain why you predict partial compensation when there is known to be variation among species in response to partner provisioning.

431-434: Here you talk about alternative hypotheses for how males will respond to changes in partner investment. I think the flow of the manuscript will be strengthened if you frame this argument in the introduction (e.g. lines 118-122). Given previous theoretical and empirical work, we know there is variation in the response rules of parents and there is a strong theoretical framework to help explain this variation. So, rather than hypothesizing that the male will either compensate or match, which is how it

reads in the introduction currently, you could mention that explanations exist for all the alternatives: no compensation, partial compensation, full compensation and matching. That would set up the discussion nicely to show that you have provided support for one of those hypotheses and eliminated the others.

Comments on Supplementary Material

Tables: For treatment and year, can you include in the table somewhere what the reference category is? (e.g. treatment: control-experimental). Otherwise it is hard to see that the experimental groups increase their provisioning relative to the control, because the effect size is negative

Page 3: *Photographing clutches in the field*

Here there is a typo in the first sentence, it should be: "Blue tit clutch photographs"

Page 9: you cite figure S5, but I believe the text is referring to figure S6.

Figure S6: I suggest including in the figure text that the numbers refer to camera distances

Concluding Remarks

Overall, this manuscript is well written, the study is original, and the results are interesting. I identified a few places where clarification is needed and provided some suggestions for how to improve the flow. However, once those comments are addressed, I feel that the manuscript will be a good contribution to our understanding of life history variation and its consequences for the evolution of bi-parental care.

Appendix B

Response to referees

Associate Editor

Board Member: 1

Comments to Author:

We have hereby received two expert reviews for this manuscript investigating hypotheses on the balance in parental care between individuals and within individuals. Both reviewers liked the experimental approach, found the results interesting, and the manuscript generally well written. However, they both also made substantial comments, that should improve the quality of the manuscript and the arguments. Reviewer 1 expressed reserve at the small sample size for the control females, and it would be good if the authors could think of a way to alleviate this, e.g. through reasoning, or statistically. The suggested additional analyses from the videos would go some way too, to make the results more robust and should additional video footage exist, it would be worthwhile to expand the number of hours observed. Additionally, both reviewers made several suggestions on how to improve the framing and interpretation of the results, that will improve the impact of the manuscript, and particularly the suggestion that the manipulation might have had unintended effects on perception of predation risk seems important - is there any relevant information in the literature to support or discount this suggestion. Both reviewers have made thoughtful and constructive suggestions that should improve the quality of the manuscript.

Thank you very much for your time and consideration of our MS. We have amended the MS to address the comments raised and provide a point by point response in bold font below each comment made in italics font.

Reviewer(s)' Comments to Author:

Referee: 1

Comments to the Author(s)

General

The main finding was very interesting and also surprising. This is because blue tits (BT) are short-lived, and one would expect from life history theory that also the females in the control group would invest heavily in rearing offspring when they have succeeded to find a suitable nest cavity (which may be in short supply) and found a mate. Therefore, I am a bit skeptical to the conclusion. This is for three reasons.

Thank you very much for your time and helpful feedback on the MS. The results were initially surprising to us also, but believe they actually make a lot of sense for two reasons. First, blue tits, whilst being short lived, are not semelparous and so still need to differentially allocate resources optimally to current versus future attempts. Second, it is intuitive to us, particularly for short-lived species with variable fecundity, that investment in egg production should partly guide anticipated brood demand and so partly determine their later provisioning rate. Note also that not all nest boxes are occupied on our sites, so survival and reproductive output are likely more important

than acquisition of a nesting opportunity.

(1) Sample size was OK for experimental females (n = 34) but rather small for controls (n = 16).

We agree that the sample size for controls is a bit low, although in line with many experimental studies of parental care in birds (see Harrison et al. 2009 JEB). Further, control nests were ‘paired’ at random with experimental nests, and so did not differ in any of the obvious confounders of provisioning investment, including: altitude; lay date; clutch size; egg volume; hatching asynchrony; hatching success; brood size; or brood age. Thus, we have no reason to believe that control versus experimental nests are systematically biased in any way, and the fact that nestling mass also differs significantly between treatment groups further points to the robustness of our provisioning results (see also below). Unfortunately, we are not able to increase the control sample size at this stage.

(2) The conclusion is based on the difference in provisioning rate, namely that it was 43% higher for experimental females than for controls.

(i) Previous studies have shown that provisioning rate in BT shows considerable variation, and thus the result may be sensitive to the small sample for controls. Provisioning rate was based on only 2h of filming once, and from these data the authors excluded the first and last 10 min of the video (to avoid effects of human visiting), leaving a relatively low amount of filming per female (100 min). Birds may forage for a while e.g. in one tree, but after some time change to a different tree of a different species. Thus, provisioning of prey items may be consistent over a time period but then change. With low sample of birds, this may cause a problem.

Our decision to base this study on 100 min of video footage is because we have found no evidence to suggest that increasing the number of observations qualitatively influences measures of provisioning investment. This is in line with a formal analysis of provisioning rates on confamilial and ecologically similar great tits which revealed that an hour of video footage during the nestling period was sufficient to capture provisioning rates (Pagani-Núñez and Senar 2013 Acta Ornithol entitled “One hour of sampling is enough: great tit *Parus major* parents feed their nestlings consistently across time”). Nevertheless, the Reviewer raises an important point that needs validating. We have consequently added the results of a repeatability analysis (using rpt-R) based on 2 x 100 min of provisioning collected on day 11-13 and again 3 days later on days 14-16. This analysis based on data collected in 2015 and 2016 as part of another experiment shows significant repeatability of provisioning rates on the two days ($r = 0.46$, 95 % CI = 0.29 - 0.61, $P < 0.001$; N = 194 provisioning periods from 55 nesting pairs), despite broods in the two intervening days being subjected to both temporary enlargement and reductions as well as an increase in chick age which is generally accompanied by increased food delivery. We have now cited the supporting studies and the repeatability analyses in the text (see lines 249-255).

(ii) With such a great difference in provisioning between female groups, and that males of experimental females had a 9% higher provisioning rate than males of control females, one would expect a strong difference in nestling body mass, but the effect was close to non-significant ($p = 0.048$, l. 335), which is very hard to explain.

Thank you for this important comment which we had not dealt with adequately in the previous version. We now make it clear on lines 62-63 and 379 that the average increase of brood provisioning rates in experimental nests was 22% (average increase in males and females combined). Second, we now make it clear on lines 387-388 and 398-401, that a 6% increase in nestling mass is in line with expectation given the 22% increase in provisioning rates of experimental pairs. This is because previous studies show that only about 20% of food intake is translated into mass gain, because not all food intake is digestible and because a substantial amount of energy intake is put into thermoregulation, maintenance and repair (Royama 1966 Ibis, Williams and Prints 1986 Condor). A 22% increase in food intake and a 20% conversion rate into mass would lead to an expected 4.5% increase in nestling mass. That we found a 6% difference in chick mass is therefore at least what would be expected given the difference in provisioning rates.

(ii) From the videos, the total number of prey items, and the proportion of green caterpillars, were recorded but not prey size. I suggest that the authors use the videos to analyse (a “blind” person) prey size to study whether or not the items were smaller in case of experimental females (which may explain the low effect on nestling mass). A way may be to compare prey size relative to beak size. Many species of caterpillars were probably involved, and different stages of development and therefore sizes.

Again, thank you for this important point that needs clarification. First, both feeding rate and prey type was coded blindly with respect to treatment (line 256). Image quality and a low frequency of seeing the bill contents makes it difficult to conduct a systematic exploration, but we have examined a number of sharp images from select videos following the Reviewer’s suggestion. All non-caterpillar prey items were about the 0.5-1 times the size of the blue tit bills (which we did not measure), so were the equivalent of 0.5-1 bill volume. In contrast, all caterpillars varied from 5-10 bill volume equivalents and so were orders of magnitude larger than other prey items, which, from what we could tell, appeared to be small spiders and insects. However, despite some variation in the size of caterpillars, the frequency of caterpillar delivery had no impact on nestling mass, presumably because they were delivered at a relatively low frequency (~10%). As such, minor differences in prey size are unlikely to explain the differences in chick growth. We have now added this information to the text on lines 365-366.

(3) The authors conclude that their finding is consistent with the hypothesis of a trade-off between breeding events (positive prenatal-postnatal investment association within a season). However, their study provides no evidence for predicted costs to the experimental females of increasing their provisioning rate, like a drop in female body mass across the nesting period, reduced subsequent survival or a reduction in reproductive success in the following year. The authors seem to have ringed all the birds, and thus it may be possible to compare female survival rate between the two groups.

We apologise for the lack of clarity and have amended the manuscript to make our point more clearly (see lines 140-147 in Introduction and 451-455 in Discussion). We started with the central assumption of life-history theory that current-future reproductive trade-offs exist, and asked the question here whether they are likely to exist between different components of reproduction within an event (i.e. between pre- and post-hatching investment)? Unfortunately, we are not in a position to conduct further analyses to test for trade-offs. First, we did not systematically capture all adults, but

ensured that at least one parent was banded to distinguish male from female (see lines 257-258). No parents were captured a second time on nest such that a change in mass is not possible to measure. Second, our return rate of adults is very low (~21%) owing to the contiguous and extensive nature of the woodlands in this area of France rendering analyses of survivorship on our sample sizes impossible (see lines 451-455).

(4) It is unknown how the removal of four eggs from the experimental females was perceived, i.e. the effect observed of provisioning may have been an “artefact” of the experimental treatment. For instance, the egg removal may have been perceived as a sign of high nest predation risk, affecting subsequent female behavior by investing more heavily when she has the opportunity (a life history argument), or to increase nestling growth rate so that the offspring would advance the time of fledging and escape nest predation. This is of course poor speculations but such possible effects of treatment may also be discussed.

Thank you for this point which we did not address adequately in the previous version and which we now do on lines 416-421 of the Discussion. It is impossible for us to verify exactly how the removal of eggs was perceived. Nests were visited once per day in both experimental and control nests such that visit rates were similar (lines 232-235) and only disappearance of an egg could generate the perception the Reviewer alludes to. However, we know of no literature that would suggest that perception of increased nest predation/failure is associated with increased provisioning. Indeed, extensive research on this subject shows that either there is no impact or more commonly, that breeders increase their reproductive investment when the perception of current success is high due to decreased predation risk (e.g. Fontaine & Martin 2006 Ecol Lett). Likewise, classic experiments in life-history trade-offs show that increased offspring predation risk (eggs in our case), should lead to a decrease in allocation to offspring investment (Reznick et al. 1990 Nature) which is the opposite to what we found. We now also provide additional information on this point on lines 274-277.

Specific points

(5) l. 56. You mention “maternal production costs”. This seems misleading because you do not seem to have recorded costs to females that are usually recorded in life history studies, namely loss of female body mass, delayed time of molting, reduced subsequent survival rate and reproduction.

We have now removed the mention of costs.

(6) l. 145. You say that it was a colour-ringer population. However, then it may be possible to measure survival rate and future reproductive success. On the other hand, you say later (l. 152) that (only) two females were found in both seasons – why so few? Annual survival rate in adult BT is usually about 40%.

As we detail above, not all parents were ringed as we only required one of the two parents to be identified to distinguish between male and female. Furthermore, the habitat our nest box population is located in is extensive contiguous woodland habitat that likely explains in part the reduced adult return rate (~21%) compared with other nest box populations which occur in more isolated habitat. As an aside, the reference to two birds was to two birds who appeared in both years of the experiment.

(7) According to life history theory, age of the bird may be important for its investment. Was there any effect of female age – was the proportion of yearlings the same in the two groups?

The Reviewer brings up a good point. In the 16 control nests we have 7 yearling females (44 %), 5 older than 1-year females and 4 NAs. In the 34 experimental nests we have 17 yearling females (50 %), 12 older than 1-year females and 5 NAs (NA's result when adults were not captured but their partner was colour-ringed). Thus, the proportion of yearlings is similar in the two treatment groups. Reviewer 1 is correct in that female age does explain significant variation in female provisioning rate. However, this does not change the effect of treatment on provisioning. However, because a fair number of individuals were of unknown age (not caught), we believe that it would be misleading to provide this additional covariate in our models which would have the additional effect of reducing the sample size. We have now added this information on female age effects (and separately male age effects, although not significant) on provisioning rates, as well as the lack of qualitative impact on treatment effects in the Supplementary Information.

(8) p. 7. Samples sizes were relatively small and so you may mention how many in each year.

We have now added the annual sample sizes per treatment group on lines 188 and 206-207.

(9). Each evening during egg laying the female may warm the eggs for a few minutes. This may improve hatching success (fighting microbes). How was the hatching success of the “under-nest-eggs”?

The Reviewer brings up another good point. As we did not mark individual eggs, we cannot say much about hatching success of the “under-nest-eggs” relative to within nest eggs. However, both treatment and control groups contained under-the-nest eggs as one under nest egg was transferred to control nests and two were retained in the experimental nests. And, almost all eggs hatched and the hatching success did not differ between treatment groups suggesting that any differences in hatching success are likely minor (see addition to Methods line 221-223 and Results lines 332-333).

(10) p. 10. You may already here in methods say that the BT is a single prey loader. You say it on l. 311 under the section of Statistical analysis.

We have now also mentioned that the blue tit is a single prey loader in the Methods (lines 258-259).

(11) Results. I suggest you add sample size for all variables/tests.

We have now made sure that sample sizes for all the variables/tests are given throughout the Methods and Results.

(12) l. 299-312 + 328-321. I suggest you present only the results here; now it is a mix of results and discussion, causes overlap/repetition with the discussion.

We actually feel that it is important for purposes of clarity to have some brief take-home messages and discussion in areas of the Results and retain those.

(13) l. 313. I suggest you give the mean percentages for the amounts of green caterpillars here, so that the readers can tell what kind of prey was available (a “good” or “bad” season). The proportion may differ substantially among years.

We have now done so (lines 365-366).

(14) l. 436. You say that BTs are socially monogamous, but this is not the case in every population.

They are socially monogamous in our population and we now make this clear (lines 504-506).

Referee 2:

Comments to the Author(s)

Summary

This study investigates the trade-off between prenatal and postnatal investment in offspring by females, and the consequences for postnatal male investment. The authors hypothesize that increased pre-natal investment by the female will result in a reduction in postnatal investment in female care, and an increase in male care to compensate. To test this hypothesis, blue tit females were experimentally induced to lay additional eggs, but to raise similar numbers of chicks as control groups, and the volume of the eggs, the male and female provisioning rates, and the weight of the nestlings were measured. Contrary to predictions, females that were induced to lay more eggs provided more postnatal care, and males did not compensate for the change in female investment. The offspring of experimental groups were also heavier, suggesting that the increase in total investment was beneficial in terms of offspring fitness. The authors discuss their results in the context of life history theory and the evolutionary maintenance of biparental care.

General Comments

This study provides a clear test of how maternal egg production costs can influence future investment in the same brood, and the consequences for partner provisioning. The methods and experimental design clearly test the hypothesis, and the manuscript as a whole is very well written. The authors present a novel test of how trade-offs between current and future reproduction operate at the scale of a breeding event, and I believe it will be of interest to the broad readership of PRSB. My comments on this manuscript are focused on some points that I believe need clarification in the methods and results, and I provide some suggestions on how to better tie together the introduction and discussion. In general, I feel that this manuscript is an important contribution to our understanding of the scale on which life history trade-offs operate, and how that influences patterns of parental care.

Thank you very much! We are grateful for your time and helpful comments.

Specific Comments

105-111: Here you provide a nice explanation for two contrasting predictions. I suggest reiterating this at the end of the introduction along with a statement of your other predictions.

Thank you, we have now added the additional text on lines 140-150.

118-122: Given that your results did not really support either compensation or matching, I suggest rewording this section. In the model you cite by Johnstone & Hinde (2006) there are contexts that result in no, partial, or full compensation, as well as matching. If you state that all are potential outcomes it will help set up the discussion of your results, particularly of no compensation or partial matching by the male.

Thank you very much for this helpful comment, and we have now done so on lines 121-126 in the Introduction.

137: Here I suggest providing an explicit statement of your hypotheses and predictions, which are not that clear from your introduction, but you explain in the discussion (e.g. lines 373-375 and lines 431-434). For example, first you ask how pre and postnatal investment will covary

and have two contrasting predictions, and second, you ask how the male responds to changes in partner effort and predict alternate outcomes based on the amount of information a male has on the value of the brood. I think being explicit here will help improve the flow between the introduction and the discussion and help justify your explanation of why your results did or did not match your predictions.

We have now added a clear statement of our predictions in the introduction (lines 140-150)

150: I suggest also including a summary of the sample sizes: the total number of nests used, the number per treatment, and the number per year, either here or at the beginning of your experimental design section.

We agree and have now added the annual sample sizes per treatment group in the Methods (lines 188 and 206-207).

152-154: Can you clarify that all other females were found only once, in either 2013 or 2014?

We have added a clarification that all other females were found only in one of the two years (lines 169-170) other than the two females mentioned in the original text. Importantly, these females received opposite treatments in the subsequent year.

207: By egg size do you mean the volume? I would state this specifically, since size could mean mass or surface area.

We have changed egg size to egg volume (lines 228 and 330).

211-212: Can you state the average number of times nests were visited?

We have added this information to the text – we visited nests nearly every day during laying and a total 8-9 times within the first ten days after the first egg had been laid. (lines 232-235)

*217: There is a typo, this should be “before **and** three days after the start of hatching”.*

Indeed, this was not clear. We have updated to: “hatching synchrony was estimated from the variance in nestling mass measured within three days after the start of hatching” (line 238-240).

258-269: It is not clear to me why total number of eggs laid, average volume of eggs laid per clutch, number of eggs incubated, number of eggs hatching and hatching synchrony were analyzed in a t-test while lay date, brood age and size, altitude, year and the proportion of caterpillars delivered were covariates in the linear model. I suggest either explaining your reasoning (e.g. to confirm that X variables did not differ between the two treatments we used a t-test, while to control for the potential confounding effects of Y variables on parental provisioning we used a linear model) or conducting some model selection procedure (e.g. stepwise regression) to determine which covariates to include.

We have now added an explanation of these two distinct statistical strategies to the text. The reviewer is correct in that the variables compared with t-tests (or non-parametric equivalents) were to ensure that experimental and control nests did not differ in some way other than the treatment effect whereas variables included in linear models were potential covariates that could differ between nests in both treatments. We have now made this clear on lines 286-290.

312-319: Here you use a t-test to compare proportion of caterpillars delivered in addition to including it as a covariate in the linear model. But in the methods (see comment on lines 258-269) you only have it as a covariate. I believe this can be easily cleared up by more explicitly stating in the methods which variables you used in each test of your hypotheses and why.

We have not added text to the methods to better explain our approach. Line about 263-265 and 299.

*342: There is a typo, this should be “male partners at experimental **nests** did not appear to reduce their provisioning rates”.*

Thank you! We have inserted the missing word (line 395).

427-429: Have there been tests of the compensation hypothesis in blue tits or a related species? If so, I suggest mentioning those studies, which could help explain why you predict partial compensation when there is known to be variation among species in response to partner provisioning.

We have now made it clear that compensation is the common response found, including for blue tits and great tits, although matching has also been shown in great tits (see lines 494-495).

431-434: Here you talk about alternative hypotheses for how males will respond to changes in partner investment. I think the flow of the manuscript will be strengthened if you frame this argument in the introduction (e.g. lines 118-122). Given previous theoretical and empirical work, we know there is variation in the response rules of parents and there is a strong theoretical framework to help explain this variation. So, rather than hypothesizing that the male will either compensate or match, which is how it reads in the introduction currently, you could mention that explanations exist for all the alternatives: no compensation, partial compensation, full compensation and matching. That would set up the discussion nicely to show that you have provided support for one of those hypotheses and eliminated the others.

Thank you, we have now made it more clear that variable responses are possible and when different ones should be expected (from line 499).

Comments on Supplementary Material

Tables: For treatment and year, can you include in the table somewhere what the reference category is? (e.g. treatment: control-experimental). Otherwise it is hard to see that the experimental groups increase their provisioning relative to the control, because the effect size is negative.

We have clarified the reference categories throughout the supplementary tables.

Page 3: *Photographing clutches in the field*

Here there is a typo in the first sentence, it should be: “Blue tit clutch photographs”

Thank you. This has been changed.

Page 9: you cite figure S5, but I believe the text is referring to figure S6.

Thank you for making us aware of this typo – we have modified the text.

Figure S6: I suggest including in the figure text that the numbers refer to camera Distances.

Thank you – we have now done this.

Concluding Remarks

Overall, this manuscript is well written, the study is original, and the results are interesting. I identified a few places where clarification is needed and provided some suggestions for how to improve the flow. However, once those comments are addressed, I feel that the manuscript will be a good contribution to our understanding of life history variation and its consequences for the evolution of bi-parental care.

Thank you for your enthusiasm and helpful comments!

Appendix B

Response to Referees

Associate Editor

Board Member

Comments to Author:

The authors did a good job on the revisions, as the reviewer testifies also. In regards to the figures, I feel that perhaps a different choice of figures to present, or one combination figure of multiple panels, would represent the results most clearly. This would include the effects on # eggs (previous figure 1), the male and female feeding rate (omitting the total, which is the sum of male and female) and the effect on nestling body mass. Such a figure would represent the whole story. Given that these are simple bar charts, a composite figure, or separate figures, should not take up too much space. For example, the axis labels could be C and E, and the bars narrower.

Thank you very much for your time and consideration of our revised manuscript. We have taken the comments on board and have now changed the figures to one combination figure with three panels as suggested.

Reviewer(s)' Comments to Author:

Referee: 1

Comments to the Author(s).

I think the authors have done a good job. Some important data/information is still missing (like female body condition and survival rate) but I understand that this is hard to get.

Thank you very much for the positive feedback.